# Enhancement of Photosynthetic Iron-Use Efficiency Is an Important Trait of *Hordeum vulgare* for Adaptation of Photosystems to Iron Deficiency

**DOI:** 10.3390/plants10020234

**Published:** 2021-01-25

**Authors:** Akihiro Saito, Shotaro Shinjo, Daiki Ito, Yuko Doi, Akira Sato, Yuna Wakabayashi, Juma Honda, Yuka Arai, Tsubasa Maeda, Takuji Ohyama, Kyoko Higuchi

**Affiliations:** Laboratory of Biochemistry in Plant Productivity, Department of Agricultural Chemistry, Tokyo University of Agriculture, Setagaya-ku, Tokyo 156-8502, Japan; hxmelt@gmail.com (S.S.); daiki.medamaoyazi@ezweb.ne.jp (D.I.); i-8yk.s4@ezweb.ne.jp (Y.D.); s.akira-0926@i.softbank.jp (A.S.); 44419021@nodai.ac.jp (Y.W.); asarihonda@yahoo.co.jp (J.H.); ar23yu@icloud.com (Y.A.); tm.nm7-28346@docomo.ne.jp (T.M.); to206474@nodai.ac.jp (T.O.); khiguchi@nodai.ac.jp (K.H.)

**Keywords:** photosynthetic iron (Fe)-use efficiency, iron deficiency, barley, sorghum, Fe economy, electron transport chain, photosystem I, photosystem II

## Abstract

Leaf iron (Fe) contents in Fe-deficiency-tolerant plants are not necessarily higher than that in Fe-deficiency-susceptible ones, suggesting an unknown mechanism involved in saving and allowing the efficient use of minimal Fe. To quantitatively evaluate the difference in Fe economy for photosynthesis, we compared the ratio of CO_2_ assimilation rate to Fe content in newly developed leaves as a novel index of photosynthetic iron-use efficiency (PIUE) among 23 different barley (*Hordeum vulgare* L.) varieties. Notably, varieties originating from areas with alkaline soil increased PIUE in response to Fe-deficiency, suggesting that PIUE enhancement is a crucial and genetically inherent trait for acclimation to Fe-deficient environments. Multivariate analyses revealed that the ability to increase PIUE was correlated with photochemical quenching (qP), which is a coefficient of light energy used in photosynthesis. Nevertheless, the maximal quantum yield of photosystem II (PSII) photochemistry, non-photochemical quenching, and quantum yield of carbon assimilation showed a relatively low correlation with PIUE. This result suggests that the ability of Fe-deficiency-tolerant varieties of barley to increase PIUE is related to optimizing the electron flow downstream of PSII, including cytochrome *b_6_f* and photosystem I.

## 1. Introduction

Photosynthesis is the most Fe (iron)-requiring process in plants [1]. The initial reaction in photosynthesis is the photosynthetic electron transfer that converts light energy into chemical energy between photosystems I and II (PSI and PSII) via cytochrome (cyt) *b_6_f* in thylakoid membranes. During this process, most of the Fe is distributed into [4Fe-4S] cluster-containing PSI and Fe-containing cyt *b_6_f* complexes and mediates light-driven electron transfer [1,2,3].

In plants suffering from Fe-deficiency, toxic reactive oxygen species are generated because of the overreduction of the plastoquinone pool due to the decrease in the Fe-containing cyt *b_6_f* and PSI function [4,5]. Fe-deficiency in plants often occurs in calcareous alkaline soils. At a high pH in calcareous soil, the equilibrium concentration of total dissolved Fe is considerably less than the concentration required for optimal growth of plants in nutrient culture [6,7]. These alkaline soils are widely distributed globally, and it is therefore crucial to understand the photosynthetic system with high Fe-use efficiency for the increase in plant biomass in such environments [5].

Of the crop plants, barley (*Hordeum vulgare* L.) is characterized as having a relatively high Fe-deficiency-tolerance. The mechanisms underlying this tolerance are coordinated in both the roots and shoots. In roots, barley has a remarkable ability to secrete mugineic acid family phytosiderophores (MAs) [8,9,10]. The MAs secreted from roots to the rhizosphere dissolve insoluble Fe by chelating and contribute significantly to the Fe-deficiency-tolerance of this plant [8]. In the shoots, barley continues growth under prolonged Fe-deficiency with less MAs and its precursor nicotianamine [11], with a similar Fe content compared to Fe-deficiency-susceptible rice [12,13], suggesting that adaptive mechanisms exist in the photosynthetic tissues of barley to cope with the Fe-deficiency. 

In the analysis of Fe-deficient barley shoots, multiple adaptive mechanisms for Fe-deficiency have been found in shoots. For instance, Fe-deficient barley demonstrates the ability to relocate Fe, N, and C effectively, accelerating the senescence of the lower leaves to compensate for the reduced photosynthesis in the young chlorotic leaves [12,14,15,16]. In chloroplasts of Fe-deficient barley, Fe is preferentially distributed into the thylakoid membranes [17]. Furthermore, prolonged Fe-deficiency (more than two weeks) induces non-photochemical quenching (NPQ), in which the excess-absorbed light energy is dissipated into heat, with the phosphorylation and monomerization of HvLhcb1, a major light-harvesting antenna protein of PSII. This NPQ induction results in protection from the photo-inhibitory effect caused by Fe-deficiency [13,18]. We propose that these features allow barley to sustain photosynthesis under long-term Fe-deficient conditions. However, the universality of these mechanisms and diversity of Fe-deficiency-tolerance in *H. vulgare* varieties is still unclear.

Barley is one of the oldest crops and has been cultivated for over 10,000 years [19]. Critical genes related to stress adaptation and domestication have been identified in recent years by new genetic analysis using the natural variation of barley [19,20,21,22]. To apply these molecular genetic approaches to study the adaptative mechanisms of photosynthesis to Fe-deficiency in barley, the appropriate quantitative traits to evaluate Fe-use efficiency are needed. 

Many studies have quantified the nitrogen (N) demand for photosynthesis among various species, using the index “photosynthetic N-use efficiency (PNUE)”. PNUE is calculated as the photosynthetic capacity per leaf N content and has been studied as an important leaf trait [23]. This index is convenient and useful because it can be determined in a small leaf area regardless of growth conditions. To date, various comparative studies of PNUE have been conducted among a wide range of plant species [24,25,26,27]. Through these studies, PNUE has been identified as an inherent trait of a species and is associated with multiple physiological factors, including CO_2_ diffusion, ribulose-1,5-bisphosphate carboxylase/oxygenase (RuBisCO) activity, and N partitioning in chloroplasts [23]. Although the concept of PNUE has been applied to other macronutrients, there is no example of applying this concept to the study of micronutrient use efficiency in plants.

In this study, we investigated the photosynthetic Fe-use efficiency (PIUE) to elucidate the difference in the Fe economy among barley varieties grown under the long-term Fe-deficient condition. We could determine PIUE appropriately by a simple method to calculate the ratio of the net CO_2_ assimilation rate to Fe content in the leaf as an index analogous to PNUE. Different barley varieties originating from regions covered with various soil pH values were selected to reveal the genetic variation in PIUE. Based on the comparison of the photosynthetic function among barley varieties with different PIUE properties, a link between PIUE and the function of the photosynthetic electron transport chain downstream of PSII to PSI is proposed.

## 2. Results

### 2.1. Fe-Deficiency-Tolerance within Barley Varieties Was Not Related to the Total Fe Content in Leaves

Barley (*H. vulgare*) is known to be more tolerant of Fe-deficiency than other crop plants. However, Fe-deficiency-tolerance within a wide range of barley subspecies has not been comparatively analyzed. In this study, we first evaluated the variation in Fe-deficiency-tolerance in *H. vulgare* species. For this purpose, four cultivars, SRB1 (Iranian), EHM1 (Japanese), ETH2 (Ethiopian), and MSS (Japanese), showing significant differences in growth under Fe-deficiency, were selected from various barley varieties (list of the full names of cultivars is in Section 4.1 and Appendix A).

Our previous study [13] had shown that long-term Fe-deficiency for more than 16 days induces the photosynthetic acclimation in the young developed leaves at fifth to sixth leaf positions. To evaluate their tolerance to the long-term Fe-deficient condition, young seedlings were transferred to the hydroponic solutions with Fe concentrations of 1–30 µM for 16 days. Then, the extent of Fe-deficient chlorosis was evaluated by analyzing the Soil Plant Analysis Development (SPAD, leaf chlorophyll meter) values in the young expanded leaves at the fifth leaf position (Figure 1). 

Among the four cultivars tested, SRB1, an Fe-deficiency-tolerant variety, did not show any chlorosis even under 1 µM Fe-deficient condition for 16 days (Figure 1A), and maintained SPAD over 50 under every Fe treatment condition (Figure 1B). Another Fe-deficiency-tolerant cultivar, EHM1, also showed no distinct Fe-deficiency chlorosis when grown in the hydroponic solution containing 3 µM Fe (Figure 1A). Under 1 µM Fe-deficient condition, EHM1 showed mild chlorosis with the SPAD value of 58% relative to control plants grown under 30 µM Fe-sufficient condition (Figure 1A,B). In contrast, Fe-deficiency-susceptible ETH2 and MSS showed clear chlorosis when grown under Fe concentrations below 3 µM (Figure 1A,B). We further confirmed that the order of Fe-deficiency-tolerance based on SPAD value and shoot biomass was SRB1 > EHM1 > ETH2 > MSS1 under more severe Fe-deficient condition (grown under hydroponic solution without Fe) (Appendix A). Hence, contrary to the shared recognition that barley is a plant species with high Fe-deficiency-tolerance, *H. vulgare* species have a considerably large variation in Fe-deficiency-tolerance, which is sufficient to study the genetic diversity of Fe-deficiency-tolerance.

In this experiment, we also evaluated the Fe content in roots and every leaf of four barley cultivars (Appendix A). However, the results showed no clear relationship between the order of Fe-deficiency-tolerance and Fe content in each tissue. Similarly, there was no regularity in the Fe distribution in all tissues of roots and shoots under Fe-deficiency within these barley subspecies (Appendix A). These results raised the new possibility that the factor responsible for Fe-deficiency-tolerance among barley varieties is not merely related to total Fe content in leaves or the distribution patterns.

### 2.2. Thylakoid Fe Concentration Was Not Correlated with Photosynthesis under Fe-Deficient Conditions

Next, we evaluate the function of photosynthetic electron transport among the four representative varieties to explore the factors affecting Fe-deficiency-tolerance among barley varieties. Because light absorption patterns and Fe content may differ significantly in leaves with different amounts of chlorophyll, we conducted this experiment by minimizing the differences in chlorophyll content among barley varieties under Fe-deficiency. For this purpose, we modified the Fe concentration in the hydroponic solution for each variety to adjust the SPAD value in the newly developed leaves (fifth to sixth leaves) under Fe-deficient conditions. The pre-cultivation and Fe treatment periods were kept constant to be the same for all varieties because the photosynthetic function of higher plants cannot be precisely evaluated without the same leaf age and the same cultivation period. This cultivation method with modified Fe concentration in medium has been a reliable method to assess the photosynthetic function in analyzing plants with different Fe-deficiency-tolerance [13,17].

After cultivation in the hydroponic solutions with 3.0 µM Fe-ethylenediaminetetraacetic acid (EDTA) for ETH2 and MSS, and 0.5 µM for SRB1 and EHM1, SPAD values of the fifth developed leaves in the four cultivars under Fe-deficiency were comparable after the 16 days of Fe-deficiency (Figure 2A). In this treatment, Fe concentrations in the young, fully expanded leaves, and the isolated thylakoid membranes, were also comparable in all cultivars under Fe-sufficient or Fe-deficient conditions (Figure 2B,C). Although all four barley varieties showed an increase in thylakoid Fe concentrations due to Fe-deficiency based on chlorophyll content, as previously reported [17], such preferential Fe distribution to the thylakoid membranes is apparently common in all varieties. 

On the other hand, the quantum efficiencies of PSII electron transport (φPSII) were lower in the Fe-deficiency-susceptible varieties (ETH2 and MSS) than in the Fe-deficiency-tolerant varieties (SRB1 and EHM1) (Figure 2D). Of note, even though the Fe-deficiency-susceptible ETH2 and MSS have the advantage of being grown in the medium containing higher Fe (3.0 µM Fe-EDTA) compared to the medium for the tolerance varieties (0.5 µM Fe-EDTA), they were still photosynthetically inferior to both SRB1 and EHM1. In other words, the slightly alleviating Fe-deficiency treatment for the susceptible varieties (ETH2 and MSS) could not improve the inferior photosynthetic function to the level of the tolerant varieties (SRB and EHM1), suggesting that Fe-deficiency-susceptible ETH2 and MSS require more Fe to maintain photosynthesis. Taken together, this result implied that there was a significant difference between these cultivars in the Fe-use efficiency for photosynthesis independent of the leaf color and amount of Fe accumulation under Fe-deficiency.

To confirm whether leaf Fe content itself is unrelated to Fe-deficiency-tolerance among barley varieties, we expanded the number of analyzed barley varieties listed in Appendix A. The newly developed leaves of 23 varieties were analyzed to determine the net CO_2_ assimilation rate and Fe concentration after 16 days of hydroponic culture with 30 µM Fe or without Fe (Appendix A). The results of the comparison of these barley varieties originating from different regions in the world revealed that there was no correlation between leaf Fe content and photosynthetic CO_2_ assimilation rate under both Fe-sufficient and Fe-deficient conditions (Figure 3A,B). As the most prominent example, MSS in the Fe-deficient condition showed almost no assimilation of CO_2_ (0.18 µmol CO_2_ m^−2^ s^−1^), although MSS had the highest Fe content (44 µmol Fe m^−2^ leaf), confirming that this Fe-deficiency-susceptible variety has very low Fe-use efficiency for photosynthesis.

### 2.3. Evaluation of ‘Photosynthetic Fe-Use Efficiency (PIUE)’

To evaluate the difference in Fe-use efficiency in photosynthesis, we compared the CO_2_ assimilation rate and Fe content in one newly developed leaf among barley varieties from a diverse geographical area (Appendix A). After the dataset was obtained, we calculated the CO_2_ assimilation rate per unit Fe content in the same leaves (Appendix A). The calculated value was denoted as photosynthetic iron-use efficiency, PIUE (Figure 4A), analogous to the index of the photosynthetic nitrogen-use efficiency, PNUE. 

Interestingly, the PIUE varied among varieties even under Fe-sufficient conditions, but the difference was more evident under Fe-deficient conditions (absolute PIUE in Figure 4A). Eight barley varieties (TW25, TRP, SRB1, AGR, GLD, EHM1, CLN, and KTN1) and the related species, *Hordeum murinum,* increased PIUE or maintained a high level of PIUE in Fe-deficient conditions compared to under Fe-sufficient conditions. In contrast, the other five barley varieties (MSS, Spont-Fin, ETH2, BNS, IGR, and MRX) exhibited a decrease in PIUE to less than half under Fe-deficiency, indicating that the photosynthetic function of these varieties seems to be highly susceptible to Fe-deficiency (Figure 4A). 

Compared to CO_2_ assimilation rate, the respiration rates in all cultivars were maintained relatively high in the Fe-deficient leaves (Appendix A), confirming that we could make measurements on active young developed leaves that had not undergone necrosis or senescence. To guarantee the reproducibility of PIUE analysis, a representative barley variety, EHM1, was grown in three batches by different experimenters at different times to obtain independent data in triplicate (Figure 4A, EHM1(1)–(3)). The relative PIUE values (not absolute PIUE values) of three different cultivation batches, EHM1(1), (2), and (3), tended to be the same with nearly identical values (Figure 4A), even though the photosynthetic rate and Fe content varied among the independent growing batches (Appendix A). This result indicates that the relative PIUE, i.e., the fold changes between the PIUE of Fe-deficient leaves and Fe-sufficient leaves, is a very reliable and specific value for each variety. 

With the aim of determining whether PIUE can quantitatively detect the Fe economy in chloroplasts among different plant species, we also investigated the PIUE among C4 graminaceous sorghum varieties. We have already reported that sorghum, unlike barley, is a plant species that cannot increase Fe allocation to thylakoid membranes under Fe-deficiency and is relatively less efficient in Fe use in the chloroplasts [17]. As expected, PIUE decreased in almost all varieties of sorghum with Fe-deficiency, except for one variety (‘#100’) (Figure 4B). Although the average PIUE value of all sorghum varieties under Fe-sufficient conditions was slightly (not significantly) higher than that of barley, the value in Fe-deficient conditions was decreased significantly to only about half the level of the barley varieties (Figure 4C). These results indicate that PIUE is a very effective indicator for quantifying differences in the Fe-deficiency responses of chloroplasts among plant species.

### 2.4. Genetic Diversity of PIUE in Barley Is the Result of Selection on Alkaline Soils

For an ecological perspective, we plotted the barley varieties in the world soil pH map created based on the Harmonized World Soil Database (HWSD) (Figure 5A). The HWSD map has been used as the most comprehensive, detailed, and updated global soil database currently available [28]. In this study, most of the barley used is cultivated and widely grown in vast areas of each country or regions. HWSD provides a raster map with 1 km spatial resolution, which is accurate enough to estimate the soil environment of the growing area. Among the eight domesticated *H. vulgare* varieties showing increased PIUE under Fe-deficiency, five varieties, TW25, TRP, SRB1, AGR, and KTN1, were originally developed in regions with alkaline soil (average soil pH above 6.5). In addition, EHM1, which is another variety with increased PIUE under Fe-deficiency, was developed in an area with relatively high soil pH (average soil pH above 6.0) in Japan (Figure 5A and Appendix A). In contrast, all varieties with low PIUE under Fe-deficiency, such as MSS, ETH2, BNS, IGR, MRX, SGH1, and ASR, originated from regions of acidic soil (average soil pH below 5.5). Therefore, a significant increase in relative PIUE can be observed in varieties developed in regions with calcareous alkaline soils such as China, Tibet, the Middle East, and the Mediterranean (Figure 5B). The same conclusion was drawn by sorting varieties based on soil pH (Figure 5C). The distribution pattern of barley varieties on the world soil map (Figure 5A) implies that soil pH may be a major environmental factor affecting the genetic variation of PIUE under Fe-deficient conditions in the development of barley cultivars.

In this study, PIUE was determined in a central part of the newly developed leaf because young leaves are most affected by Fe-deficiency and the position inducing the long-term acclimation mechanisms to Fe-deficiency [13]. Therefore, it was necessary to confirm whether the analysis of local PIUE in young leaves could explain the differences in total growth under Fe-deficiency in barley varieties. For this purpose, the shoot and root biomass of the 18 representative barley varieties was compared with their relative PIUE value as the ratio of Fe-deficient to Fe-sufficient conditions. There was a significant positive correlation between relative PIUE and the relative biomass of the shoot (Figure 6A) or root (Figure 6B). This result is reasonable because the upper young leaves have a significant role as the plant canopy, which is the most favorable position for CO_2_ assimilation and plant growth [30]. Therefore, we confirmed that the analytical method of PIUE is reliable and not only shows Fe-use efficiency in the analyzed young leaf but also reflects the growth of the whole plant under Fe-deficient conditions. 

### 2.5. Photosynthetic Properties Associated with the Relative PIUE Value in Barley Varieties

To estimate which photosynthetic processes are responsible for the differences in PIUE under Fe-deficient condition, we compared various indexes related to photosynthesis: the maximal quantum yield of PSII photochemistry (Fv/Fm) as an index for PSII integrity, the quantum yield of PSII photochemistry (Fv’/Fm’), photochemical quenching (qP), and effective quantum yield of electron transport (φPSII) for functions of photosynthetic electron transport, non-photochemical quenching (qN and NPQ) for thermal dissipation, stomatal conductance for CO_2_ gas diffusion (g_s_), and φCO_2_ for the efficiency of CO_2_ assimilation. All analyses were performed simultaneously with the measurement of CO_2_ assimilation analysis in the same plants used for Figure 3, Figure 4, Figure 5 and Figure 6. The newly developed leaves of 18 barley varieties (see the caption in Appendix A) were analyzed at a moderate light strength of 500 μmol photons m^−2^ s^−1^. We conducted correlation analysis and principal component analysis (PCA) using the various photosynthetic data (Appendix A). The datasets for these multivariate analyses consisted of log-transformed relative values (−Fe/+Fe) of photosynthetic parameters and other measured values (Appendix A). 

The results of the correlation analysis showed that PIUE was positively correlated with the value of qP (r = 0.843), followed by the other parameters related to the electron transport within PSII, such as φPSII (r = 0.835) and Fv’/Fm’ (r = 0.739) (Figure 7A). On the other hand, g_s_, Fv/Fm, and φCO_2_ had a lower correlation (r = 0.727, 0.685, and 0.548, respectively) with PIUE than the above parameters (Figure 7A). NPQ had little or no correlation with PIUE (Figure 7A). Respiratory activity (Resp) was maintained in all cultivars under Fe-deficiency and thus showed no relationship to the pattern of PIUE changes among barley varieties. SPAD value and leaf Fe content are commonly used as indicators of Fe-deficiency-tolerance, but their association with PIUE within barley cultivars was lower than that of the parameters related to the electron transport. A heatmap was drawn to visualize the correlation matrix between variables related to photosynthesis (Figure 7B), confirming that qP and PIUE were most associated with each other as they were in the same branch of the hierarchical clustering.

From PCA analysis, the first principal component (PC1) and second principal component (PC2) were used in biplots (Figure 8) because the sum of the contribution of PC1 and PC2 was sufficiently large to explain the variance (63% and 18%, respectively). The loading vector of PIUE showed a direction similar to that of qP, forming the same cluster. However, Fv/Fm, Fv’/Fm,’ φPSII, and φCO_2_ formed a different cluster, which was slightly separated from PIUE and qP (Figure 8). NPQ and qN showed different directional vectors from the PIUE. Leaf Fe concentrations were entirely unrelated to all photosynthesis-related parameters, confirming that leaf Fe content is not an indicator of photosynthesis within barley varieties (Figure 3). These results suggest that different responses of PIUE to Fe-deficiency within barley varieties were most related to the electron transfer system represented by qP. The degree of PSII integrity, photoprotective thermal dissipation, and carbon assimilation enzymes in stroma could not be the primary factors associated with PIUE because of the low relevance of Fv/Fm, NPQ, and φCO_2_ to PIUE.

## 3. Discussion

### 3.1. PIUE, as a New Indicator that Defines Photosynthetic Fe-Use Efficiency

In this study, we introduced a novel indicator, PIUE, to differentiate Fe economy during photosynthesis under Fe-deficient conditions. This attempt successfully characterized the Fe-use efficiencies in photosynthesis of *H. vulgare* varieties (Figure 4A and Figure 5A). As far as we are aware, this is the first study assessing the photosynthetic Fe-use efficiency as a quantitative trait. Interestingly, PIUE was associated with qP when compared among barley varieties (Figure 7 and Figure 8). The qP is literally the index of the ratio of energy quenched for photosynthesis [31] and largely reflects the redox state of Q_A_ and downstream of PSII, including the PSI and Calvin cycles [31,32,33]. In the present study, PSII integrity, as indicated by Fv/Fm, was maintained in almost all barley varieties (Appendix A). Stomatal conductance and φCO_2_, which reflect the activity of CO_2_ diffusion and Calvin cycle, showed a relatively low correlation with PIUE (Figure 7). Therefore, the reason for the difference in qP among barley varieties appears to be restricted downstream of PSII, such as in cyt *b_6_f* and PSI (Figure 9A). Because cyt *b_6_f* and PSI are the most Fe-requiring photosynthetic apparatus and the primary sufferer of Fe-deficiency in plants [1,3,5], the postulation that the key mechanism for increased PIUE under Fe-deficiency to exist downstream of PSII seems reasonable. 

### 3.2. Possible Mechanisms that Affect PIUE

There was no difference in Fe content on thylakoid membranes between barley varieties in this study (Figure 2C). Therefore, the reorganization of thylakoid proteins and optimization of Fe distribution to various Fe-containing proteins may also be important to increase or maintain PIUE under Fe-deficient conditions. Since cyt *b_6_f* and PSI are the complexes that require the most Fe in plant cells [1,3,5] and are highly related to the qP, there are possibilities that some of the Fe-transport/delivery pathways to these complexes may be involved in the efficiency of Fe-use in barley. A significant decrease of qP without alteration of the PSII complex has been reported in the mutants lacking HCF101 and APO1, the [4Fe-4S] cluster delivering proteins for PSI [31,34]. Similarly, CRR2, CRR6, PGR3, and PGR5 proteins for the cyclic electron transport influenced the qP and optimized the electron transport around cyt *b_6_f* and PSI [35,36,37]. It is reasonable to speculate that related pathways may contribute to the Fe-use efficiency in Fe-deficient barley. 

The increase in accumulation of other transition metals such as Mn and Cu in leaves under Fe-deficiency is a long-known phenomenon [38]. We have also previously investigated the increased accumulation of various elements in Fe-deficient barley [16]. In this study, among the barley varieties, we found a marked increase in Mn accumulation in Fe-deficient leaves of the susceptible varieties, MSS, ETH2, SGH1, and Sponta-Fin (Appendix A). Excess Mn is also known to decrease PSI content [39]. In Arabidopsis, vacuolar Mn transporter MTP8 confers the tolerance to Fe-deficiency chlorosis by preventing interference of Mn in Fe acquisition [40]. Although it is unclear whether the excess Mn accumulation found in the Fe-deficiency-susceptible varieties is the cause or the consequence of the severe Fe-deficiency symptoms, excess Mn could reduce the Fe-use efficiency in chloroplasts. In addition to the Fe distribution mechanism in the chloroplast, the optimizing mechanisms of balance between Fe and other elements will need to be considered.

### 3.3. Association of PIUE with Other Fe-Deficiency Acclimation Mechanisms Previously Found in Barley

The increase in NPQ under Fe-deficient condition has been found in Fe-deficient barley by comparing photosynthetic function between barley and rice [13]. The elevated NPQ, namely thermal dissipation of light energy, is the reason why barley is less susceptible to photo-oxidative stress under Fe-deficiency than rice. The critical factor in this mechanism has been identified as isoforms of the HvLhcb1 [13,18]. This study confirmed that all barley varieties tested here significantly increased NPQ under Fe-deficient condition (Appendix A). As the NPQ induction is a primary photoprotective mechanism for PSII, almost all barley cultivars indeed maintain high Fv/Fm (0.7–0.8) under Fe-deficient conditions (Appendix A). This property is entirely different from rice and other Fe-deficiency-susceptible plants, where Fe-deficiency damages PSII. Thus, it becomes clear that the NPQ induction to protect PSII is a universal (fundamental) mechanism among barley varieties (Figure 9B).

Like NPQ induction, an increase in Fe supply to thylakoid membranes was observed in all analyzed barley varieties with different Fe-deficiency-tolerances (Figure 2C). This observation also suggests that the preferential supply of Fe to thylakoid membranes appear to be universal strategies to cope with Fe-deficiency in *H. vulgare* species (Figure 9B). Because of this universality, both NPQ and Fe were not correlated with PIUE (Figure 7A and Figure 8). Therefore, these two universal mechanisms may be independent of the mechanism of PIUE induction under Fe-deficiency within *H. vulgare* species (Figure 9B). Comparisons between barley varieties with different values for PIUE could help elucidate the unknown Fe-deficiency adaptation mechanism to save Fe in chloroplasts, which is not universal within barley varieties.

Of note, however, rice and *Arabidopsis* could not induce the light-harvesting complex II (LHCII)-mediated NPQ under Fe-deficiency [3,13], suggesting that the photoprotective mechanism is not universal in the plant kingdom. Similarly, sorghum shows no sign to increase Fe in thylakoid membranes under Fe-deficiency [17]. Therefore, it is also evident that this characteristic adaptive mechanism commonly found among barley varieties is the key process in the greater tolerance of Fe-deficiency than in other plant species. In other words, cooperation with this species-specific (NPQ-induction and preferential Fe supply into thylakoids) and the varieties-specific mechanism (PIUE induction) would increase the total Fe-deficient-tolerance level in plants.

### 3.4. New Insights on Barley Cultivation in Respect of Fe Nutrition 

Recent progress in genome analysis of barley has revealed deep links between geographic distribution and genetic diversity in this species [41]. Assuming that the genetic diversity of PIUE changes depending on soil pH, it is possible to explain the current global distribution of barley varieties with different PIUE changes (Figure 5). Among the barley cultivars used in this study, six varieties with increased PIUE (TW25, TRP, SRB1, AGR, KTN1, and EHM1) originated from regions of alkaline soil (average soil pH above 6.5) or relatively high pH compared to the surrounding area (Figure 5A). 

Areas with an average soil pH greater than 6.3 contain a wide range of alkaline soils because the actual soil pH is heterogeneous, as shown in Appendix A (the column “Soil pH range at origin”). In aerated soils like farmlands for barley cultivation, the solubility of Fe decreases by a factor of 1000 for each unit increase in pH [7]. Therefore, barley grown in these areas is at high risk of low Fe availability. As we have reported [42], even a small change in pH significantly impacted the elongation of barley roots, indicating that such a slight increase of soil pH could alter the energy metabolism. In the case of *Ambrosia artemisiifolia*, pH 7 is a serious environment, inhibiting the formation of flowers and pollen and delaying their growth [43]. Therefore, plants grown in an area with an average soil pH exceeding 6.5 must have a flexible mechanism to acclimate to Fe-deficiency to obtain sufficient yields. In this context, because the increase in PIUE would be a desirable trait for cultivation in alkaline soils, it is likely that these physiological traits have been inadvertently selected for in these regions during the long history of barley cultivation.

Conversely, varieties with a PIUE decreased by Fe-deficiency (MSS, BNS, IGR, Sponta-Fin, and ETH2) originated from acidic soil regions (Figure 5A). In such acidic soil environments, plants have almost no opportunity to encounter Fe-deficiency, but rather, stress from excess Fe will occur in these plants. Thus, unlike the case in alkaline soils, the ability to induce high PIUE provides no profit in obtaining high yields in acidic soil environments. Therefore, it is conceivable that part of the barley varieties adapted to acidic soil environments lost the ability, or the environment did not provide selective pressure to obtain the ability to induce high PIUE under Fe-deficiency.

Two varieties, CLN and GLD, did not match the above scenarios. Both cultivars increased their PIUE by Fe-deficiency despite growing in acidic soil (Figure 5). CLN was produced in 1938 in the USA by crossing two varieties, ‘Davidson’ and ‘Sunrise’. Of these, ‘Sunrise’ was selected from the Japanese semi-dwarf variety ‘Nakano Wase’. GLD was originally a dwarf mutant created in 1967 by gamma-ray irradiation of ‘Maythorpe’, a hybrid of the Danish ‘Maja’ and British ‘Goldthorpe’. Both CLN and GLD have a dwarf mutation in their genome. However, there was no evident correlation between PIUE and dwarf morphology in barley (Appendix B, Figure A1). Therefore, it is more plausible to consider that these “new varieties” are no longer related to the soil pH of the current growth area because of the complex breeding process in the 20th century. 

### 3.5. The Origin of the Trait of PIUE Increase 

The question is raised as to when barley obtained this property of PIUE induction in response to Fe-deficiency. Wild barley, *H. vulgare* spontaneum, an ancestral species of cultured barley, is currently distributed globally. In this study, we found that the spontaneum originating from Iran (average soil pH above 7.0) had a higher PIUE under Fe-deficiency than that from Finland (average soil pH < 5.5) (Figure 4A). Furthermore, another wild-barley, *Hordeum murinum*, originating from Spain (average soil pH ~7.0) and closely related to *Hordeum vulgare*, has the highest absolute PIUE among plants used in this study (Figure 4A). These results suggest that the diversity of PIUE within the *Hordeum* species may exist before cultivation. Afterward, the beneficial genes linked to PIUE induction under Fe-deficiency would be integrated through human selection, resulting in the present Fe-efficient barley varieties. To elucidate this, it would be worth investigating the PIUE in various plant species. The relationship between PIUE and the adaptation of plants to Fe-deficiency will be a subject of future studies.

### 3.6. C4 Photosynthesis Is Disadvantageous in PIUE Increase under Fe Deficient Conditions 

PNUE in the C4 photosynthetic plant is generally greater than that in the C3 plant [24]. In addition, C4 plants are tolerant to drought stress [44] and have higher photosynthetic water-use efficiency (PWUE) than C3 plants [45]. However, unlike PNUE and PWUE, our results clearly showed that the average PIUE value collected from sorghum varieties (which have C4 photosynthesis) was significantly decreased by Fe-deficiency (Figure 4C). This result suggests that C4 photosynthesis is a disadvantage in acclimatizing to Fe-deficiency (Figure 4B,C).

Indeed, C4 plants require a larger amount of Fe in the vascular bundle sheath cells that accumulate PSI complexes more than C3 plants [46]. Our previous studies demonstrated that sorghum has a lower ability for Fe distribution to the thylakoid membranes under Fe-deficient conditions than barley [17]. Only one sorghum variety, ‘#100′, seemed to maintain the PIUE under Fe-deficient conditions. It may be worthwhile to study whether there is an optimizing mechanism for PIUE in sorghum varieties. 

### 3.7. The Future Application of PIUE

Fe takes various chemical forms in vivo and it is therefore difficult to quantitatively identify the precise positions and chemical structures of trace amounts of Fe. This difficulty has made it challenging to perform comprehensive screening of plants with different Fe-saving systems in chloroplasts. In this regard, PIUE has an advantage in that it can be analyzed in any leaf position in any plant species (Figure 4) in both laboratory and field environments. Such applications of PIUE will further deepen our knowledge of the properties of Fe-economy in chloroplasts.

Quantitative trait loci (QTLs) analyses and genome-wide association studies (GWAS) have been performed using indices related to resource-use efficiencies. Recently, multiple QTLs or genes related to such efficiencies in photosynthesis have been identified [47,48]. The classification of plants by relative PIUE (−Fe/+Fe) can produce apparently different results compared to the traditional indices, SPAD, and Fe content (Figure 7). We expect that PIUE may become a novel index as a quantitative trait for such genetic analyses. For this purpose, F_2_ crossing lines from several barley varieties with different PIUE responses are now under development.

In summary, photosynthetic Fe-use efficiency can be a key trait in maintaining growth under Fe-deficiency within *Hordeum* species (Figure 6). The difference in Fe-deficiency-tolerance within *H. vulgare* species is associated with its ability to increase PIUE under Fe-deficiency. It is likely that the mechanism of the PIUE increase in Fe-deficient barley is related to optimizing the electron flow downstream of PSII. Further analyses of Fe partitioning and the compositional changes of thylakoid proteins will be needed to clarify this assumption. With the application of PIUE, it is expected that molecular mechanisms will be identified to improve Fe-economy under Fe-deficient conditions.

## 4. Materials and Methods

### 4.1. Plant Materials and Growth Conditions

Varieties of barley (*Hordeum vulgare* L. ‘Ehime Hadaka 1’ (EHM1), ‘Shiro Hadaka 1’ (SRH1), ‘Kairyo Ogara’ (KRO), ‘Haruna Nijo’ (HRN), ‘Akashinriki’ (ASR), ‘Saga Hadaka 1’ (SGH1), ‘Musashinomugi’ (MSS), ‘Colonial’ (CLN), ‘Bowman’ (BWM), ‘Bowman near-isogenic line uzu1.a’ (BWM-uzu), ‘Bonus (BNS), ‘Igri’ (IGR), ‘Barke’ (BRK), ‘Morex’ (MRX), ‘Golden Promise’ (GLD), ‘Tripoli’ (TRP), ‘Sarab 1’ (SRB1), ‘Katana 1’ (KTN1), ‘Tibet White 25’ (TW25), ‘Ethiopia 2’ (ETH2), ‘Agriochriton’ (AGR), ‘Spontaneum’ originated from Iran (Sponta-Iran), ‘Spontaneum’ originated from Finland (Sponta-Fin),’ and *Hordeum murinum* L.(Murinum)) and sorghum (*Sorghum bicolor* L. spp. ‘#2’, ‘#20’, ‘#48’, ‘#49’, ‘#95’, ‘#100’, and ‘#168’) used in this study are summarized in Appendix A. Among them, 15 barley varieties were kindly provided by Professor Kazuhiro Satoh (Barley Germplasm Center, Okayama University, Japan) and the other eight barley accessions, including wild species of barley, were kindly provided by the Nordic Genetic Resource Center (https://www.nordgen.org/en/plants/plant-material/). ‘BWM-uzu’ was also kindly provided by the National Small Grains Collection, at the United States Department of Agriculture (https://www.ars-grin.gov/npgs/collections.html). All sorghum varieties were provided by EARTHNOTE Co., Ltd., Tokushima, Japan. Seeds of barley and sorghum were germinated on moist filter paper after water absorption at 4 °C for several days. 

The seedlings were then grown hydroponically in a growth chamber at 24 °C. The growth light intensity was set at 200–300 μmol photons m^−2^ s^−1^ under 14 h light/10 h dark cycles. After germination, pre-culture was carried out until the seed nutrients were completely consumed. Pre-culture was conducted with a half-strength hydroponic solution containing 30 μM Fe-EDTA. After 11–12 days of the pre-culture, the length of the second leaf exceeded that of the first leaf, then the plants were transferred to the hydroponic solution (0.7 mM K_2_SO_4_, 0.1 mM KCl, 0.1 mM KH_2_PO_4_, 2 mM Ca(NO_3_)_2_, 0.5 mM MgSO_4_, 10 µM H_3_BO_3_, 0.5 µM MnSO_4_, 0.5 µM ZnSO_4_, and 0.2 µM CuSO_4_ and (NH_4_)6Mo_7_O_24_) with 30 µM Fe-EDTA for barley and 100 µM Fe-EDTA for sorghum to provide Fe-sufficient conditions. No Fe was added after pre-culture to induce Fe-deficiency for all barley varieties, except for the experiment in Figure 2. To prepare Fe-deficient sorghum varieties, 3–10 µM of Fe-EDTA was added to maintain their growth at a SPAD level of approximately 15.

For the experiments in Figure 2, Fe concentrations of the hydroponic solution for the Fe-deficient treatment were adjusted from 0 to 1 μM for the Fe-efficient varieties (EHM1 and SRB1) and from 2 to 3 μM for the Fe-inefficient varieties (MSS and ETH2) to standardize the chlorophyll content across all the varieties analyzed. Since the current study focused on the photosynthetic acclimation to Fe-deficiency in young developed leaves [13], the mechanism is fully induced after 16 days of Fe-deficiency. Therefore, the Fe treatment was conducted for 16–20 days after the pre-culture in the case of barley and for 10–14 days in the case of sorghum. The nutrient solutions were renewed every 2–4 days. For varieties suffering from excessive heavy metal stress under Fe-deficient conditions, the concentrations in the hydroponic solution of Mn, Zn, and Cu were reduced to 0.1, 0.1, and 0.05 µM, respectively. The details are described in Appendix A. 

### 4.2. Isolation of Thylakoid Membranes

Young, fully expanded chlorotic leaves and corresponding control leaves were analyzed after 2–3 weeks when the chlorosis symptoms remained constant. Thylakoid membrane extraction was simplified as described previously [49]. Reagents and the detailed procedures are presented in Appendix B (Methods A1). In brief, leaf strips were mildly homogenized with an isotonic solution and crude chloroplasts were collected in the flow-through of Miracloth (Merck, Burlington, MA, USA). Crude chloroplasts were ruptured by a hypotonic solution, and the thylakoid fraction was obtained. The thylakoid membrane was washed with EDTA solution [50]. 

### 4.3. Measurement of Chlorophyll and Fe Content in Leaves

The SPAD value (index of total chlorophyll content in a leaf area) of the central parts of the youngest, fully developed leaves (5th to 6th leaves for barley, 4th to 5th leaves for sorghum—the cotyledon was counted as the first leaf) was analyzed using SPAD-502 (Minolta, Osaka, Japan). Chlorophylls were quantified after extraction with 80% (*v*/*v*) acetone [51]. 

Fe concentration was measured using an atomic absorption spectrometer (AA-6300, Shimadzu, Tokyo, Japan) coupled with a graphite furnace atomizer (GFA-EX7i, Shimadzu) after the digestion of leaf materials, as described previously [13]. To determine trace Fe in thylakoid membranes (Figure 2C), extremely clean reagents, instruments, and atmosphere were prepared. Digestion was performed with Teflon tubes (perfluoroalkoxy alkanes (PFA) sample test tube, T43-012, Matsuura-Seisakusho Co. Ltd., Tokushima, Japan), which release no Fe when heated with HNO_3_. All Teflon tubes and plastic tips with filters were soaked in diluted HNO_3_ before use. The bench was covered with a simple household greenhouse composed of steel pipes and a plastic cloth. 

### 4.4. Measurement of Photosynthesis

Leaf gas exchange measurements were coupled with measurements of chlorophyll fluorescence using an open gas exchange system (LI-6400XT, LI-COR, Inc., Lincoln, NE, USA) with an integrated fluorescence chamber head (LI-6400-40 leaf chamber fluorometer; LI-COR, Inc.). Youngest, fully developed leaves (5th to 6th leaves for barley, 4th to 5th leaves for sorghum—the cotyledon was counted as the first leaf) were analyzed, and CO_2_ assimilation rate (*A*: μmol CO_2_ m^−2^ s^−1^) and stomatal conductance (g_s_: mol H_2_O m^−2^ s^−1^) were measured at 500 μmol photons m^−2^ s^−1^. Respiration rate (Resp: μmol CO_2_ m^−2^ s^−1^) was measured as the amount of CO_2_ released in darkness. Measurements were set as follows: block temperature = 24 °C, reference CO_2_ concentration = 400 μmol m^−2^ s^−1^, flow = 500 μmol s^−1^, and actinic light intensity = 500 μmol photons m^−2^ s^−1^ (containing 10% of blue light). The quantum yield of carbon assimilation (φCO_2_) was calculated as (1):φCO_2_ = *A*_light_ − *A*_dark_/*I* α_leaf_,(1)
where *A*_light_ and *A*_dark_ are the CO_2_ assimilation rate in light and dark respectively, *I* is the incident photon flux density, and α_leaf_ is the leaf absorptance.

Chlorophyll fluorescence analysis was performed for 18 barley varieties, as shown in Figure 7, simultaneously with gas exchange analysis using LI-6400XT and LI6400-40 (LI-COR, Inc.), as previously described [13]. In brief, the maximum level of fluorescence was determined after receiving a saturating pulse in a dark-adapted state (Fm) or in the light-acclimated state (Fm’). The minimum fluorescence (Fo) was measured using a sufficiently low intensity. After the steady-state level (Fs) of the actinic irradiance was measured, the actinic light was turned off, and Fo’ (the minimum fluorescence level in the light-acclimated state) was determined with far-red light. Calculations were performed using the following parameters: Fv/Fm = (Fm − Fo)/Fm(2)
where Fv/Fm is the maximal quantum yield of PSII photochemistry,
Fv’/Fm’ = (Fm’ − Fo’)/Fm’(3)
where Fv’/Fm’ is the quantum yield of PSII photochemistry,
φPSII = (Fm’ − Fs’)/Fm’(4)
where φPSII is the effective quantum yield of electron transport in the light-acclimated state,
qP = (Fm’ − Fs)/(Fm’ − Fo’)(5)
where qP is the photochemical quenching coefficient,
qN = (Fm − Fm’)/(Fm‘ − Fo’)(6)
where qN is the non-photochemical quenching and
NPQ = (Fm − Fm’)/Fm’(7)
where NPQ is also the non-photochemical quenching.

### 4.5. Calculation of PIUE

After the measurements of the CO_2_ assimilation rate (μmol CO_2_ m^−2^ s^−1^), the central part of the analyzed leaf was cut out with a razor, washed with distilled water, and the area of the detached leaves was calculated using a scanning image based on the area value of the grid paper. The dry weight of the detached leaves was measured after overnight drying at 80 °C. The Fe concentration per leaf area (μmol Fe per m^2^ leaf area) was calculated based on the dry weight (DW) per m^2^ (g leaf DW per m^2^ leaf area). Finally, PIUE (μmol CO_2_ mol^−1^ Fe s^−1^) was calculated as the CO_2_ assimilation rate divided by the Fe concentration per leaf area. To guarantee the reproducibility of PIUE analysis, a representative barley variety, ‘Ehime Hadaka 1’ (EHM1), was grown thrice to obtain three repetitive datasets. 

### 4.6. Statistical Data Analysis

The log-transformed value of Fe-deficiency (−Fe)/Fe-sufficiency (+Fe) of each index of chlorophyll fluorescence, stomatal conductance, CO_2_ assimilation rate, biomass, and leaf Fe concentration were used to evaluate the variation in the photosynthetic traits among 18 barley varieties grown under Fe-sufficient and Fe-deficient conditions (Appendix A). Clustering heatmap analysis, scatter plot matrices, and principal component analysis (PCA) were performed using R original packages, pairs.panels and corrplot, respectively (R version 4.0.3). 

## Figures and Tables

**Figure 1 plants-10-00234-f001:**
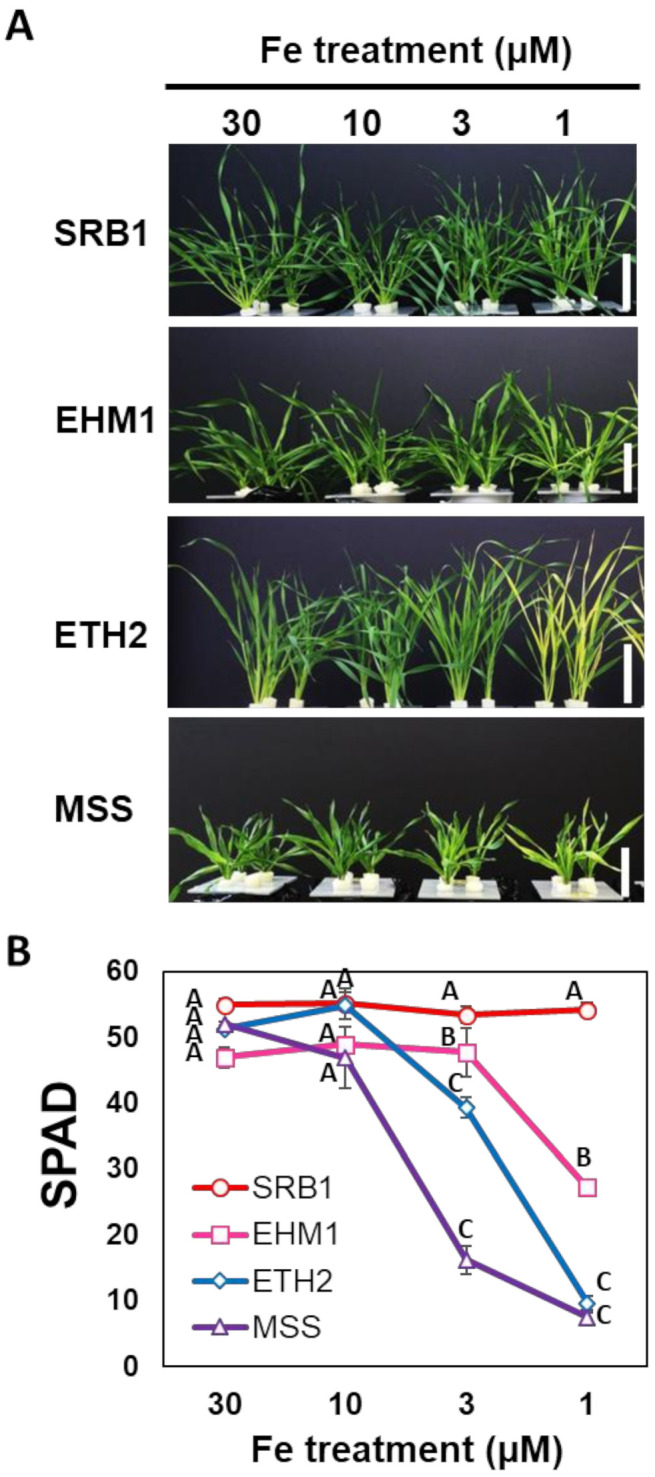
Fe-deficiency-tolerance among four representative barley varieties. Photographs (**A**) and SPAD (chlorophyll meter) values of young developed leaves (**B**) of four barley varieties grown under different Fe treatment (30, 10, 3, and 1 µM Fe) for 16 days. Before the Fe treatments, seedlings were grown with normal Fe concentration (30 µM Fe) for 11–12 days as a pre-culture until the seed nutrition was consumed. Significant differences among different varieties were tested using the Tukey’s multiple test at each Fe treatment (N = 3 ± standard error, *p* < 0.05, same letters indicate no significant difference). ‘Sarab 1′ (SRB1) and ‘Ehime Hadaka 1′ (EHM1) are the tolerant cultivars and ‘Ethiopia 2′ (ETH2) and ‘Musashinomugi’ (MSS) are the susceptible barley cultivars.

**Figure 2 plants-10-00234-f002:**
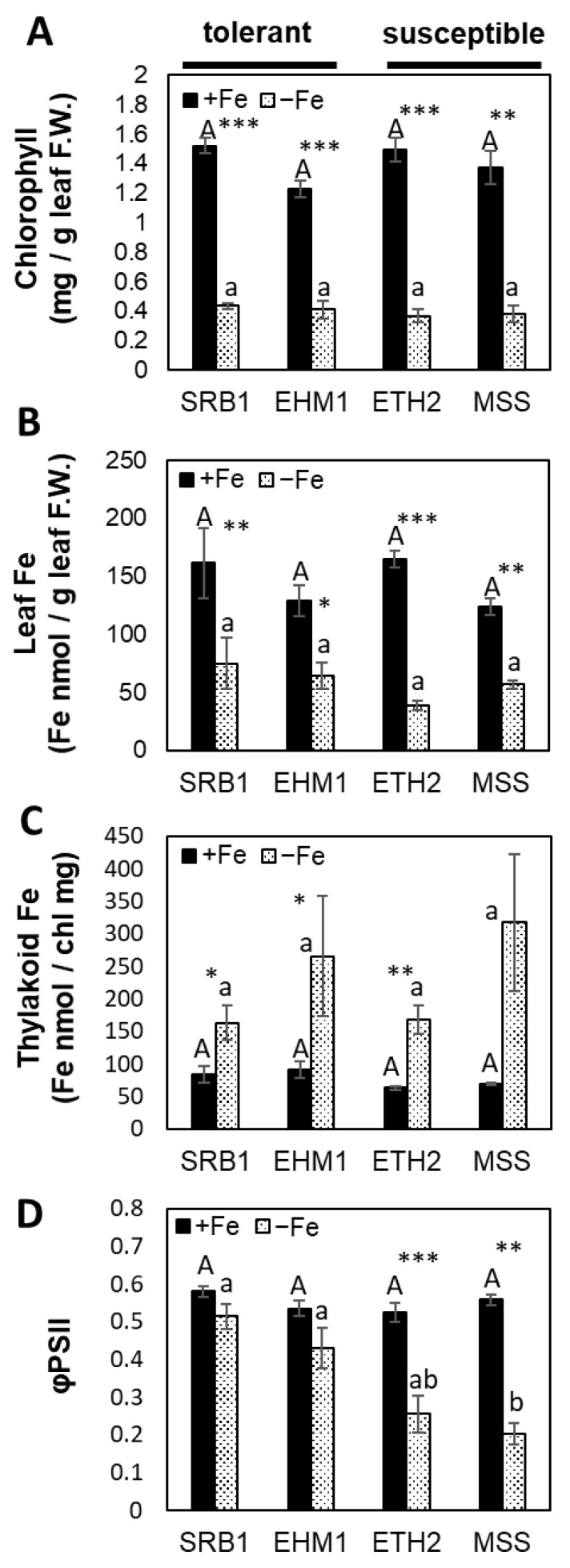
Comparison of the physiological traits among four representative barley varieties. Data were obtained from young developed leaves (5–6th leaves) of each variety. Chlorophyll concentrations (**A**), Fe concentrations in leaf (**B**), Fe concentrations in the thylakoid membranes (**C**), and chlorophyll fluorescence parameter for the quantum efficiencies of PSII electron transport (φPSII) (**D**) of four barley varieties grown under Fe-sufficient (+Fe) and mild Fe-deficient (−Fe) conditions for 16–20 days are shown. Fe concentration in the −Fe solution was adjusted to 3.0 µM Fe for ETH2 and MSS, and 0.5 µM Fe for SRB1 and EHM1 to prepare similar chlorotic leaves at the fifth leaf position in all four cultivars. The significant differences between Fe treatments were determined using the Student’s *t*-test ((N = 3 ± standard error, * *p* < 0.05, ** *p* < 0.01, *** *p* < 0.001), and between-varieties Tukey’s multiple test was used (*p* < 0.05, no significant difference between the same lowercase or between the same capital letters).

**Figure 3 plants-10-00234-f003:**
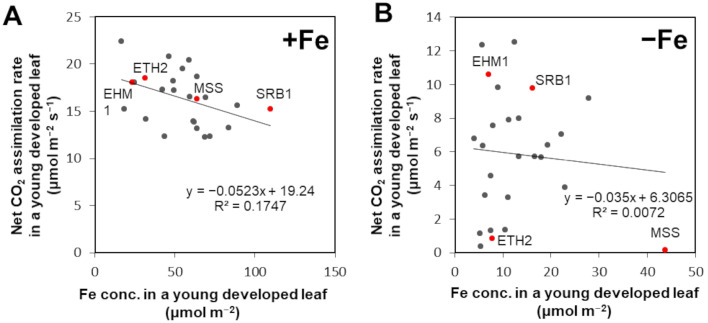
Correlation between leaf Fe concentration and net CO_2_ assimilation rate among barley varieties. Correlation data among 23 barley varieties grown under Fe-sufficient (+Fe) condition (**A**) or Fe-deficient (−Fe) condition (**B**). The original data on barley (including *H. vulgare* spontaneum and *Hordeum murinum*) used for the analysis are presented in Appendix A. Significance of differences in Pearson’s correlation coefficient between treatment groups was determined using the Student’s *t*-test (*p* < 0.05, no notation for those with no significant difference). ‘Sarab 1’ (SRB1) and ‘Ehime Hadaka 1’ (EHM1) are the tolerant cultivars, and ‘Ethiopia 2’ (ETH2) and ‘Musashinomugi’ (MSS) are the susceptible barley cultivars.

**Figure 4 plants-10-00234-f004:**
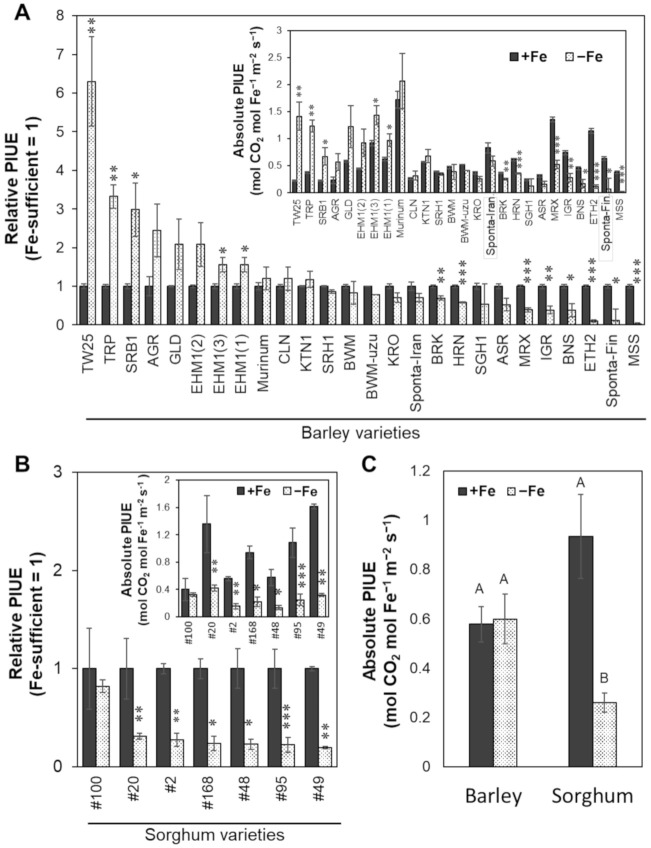
Variation in the photosynthetic Fe-use efficiency (PIUE) among barley and sorghum varieties. The relative value of PIUE of the Fe-deficient (−Fe) leaves to the Fe-sufficient (+Fe) leaves of barley (**A**) and sorghum (**B**) are shown. Inserted figures in (**A**,**B**) are absolute value of PIUE, representing molar amounts of CO_2_ that can be assimilated per mole leaf Fe in one second of time. Comparison of an average of absolute PIUE values for all barley and all sorghum varieties (**C**). The significant differences between Fe-sufficient and Fe-deficient plants in (**A**,**B**) were tested using the Student’s *t*-test (N = 3 ± standard error, * *p* < 0.05, ** *p* < 0.01, *** *p* < 0.001), and among the data of sorghum and barley in (**C**) were tested using the Tukey’s multiple test (N = 3 ± standard error, *p* < 0.05, same capital letters indicate no significant difference). Abbreviations of plant variety names are summarized below: ‘Ehime Hadaka 1’ (EHM1), ‘Shiro Hadaka 1’ (SRH1), ‘Kairyo Ogara’ (KRO), ‘Haruna Nijo’ (HRN), ‘Akashinriki’ (ASR), ‘Saga Hadaka 1’ (SGH1), ‘Musashinomugi’ (MSS), ‘Colonial’ (CLN), ‘Bowman’ (BWM), ‘Bowman near-isogenic line uzu1.a’ (BWM-uzu), ‘Bonus (BNS), ‘Igri’ (IGR), ‘Barke’ (BRK), ‘Morex’ (MRX), ‘Golden Promise’ (GLD), ‘Tripoli’ (TRP), ‘Sarab 1’ (SRB1), ‘Katana 1’ (KTN1), ‘Tibet White 25’ (TW25), ‘Ethiopia 2’ (ETH2), ‘Agriochriton’ (AGR), ‘Spontaneum’ from Iran (Sponta-Iran), ‘Spontaneum’ from Finland (Sponta-Fin), and *Hordeum murinum* L. (Murinum). Triplicate data of EHM1 are denoted as (EHM1(1), EHM1(2), and EHM1(3)).

**Figure 5 plants-10-00234-f005:**
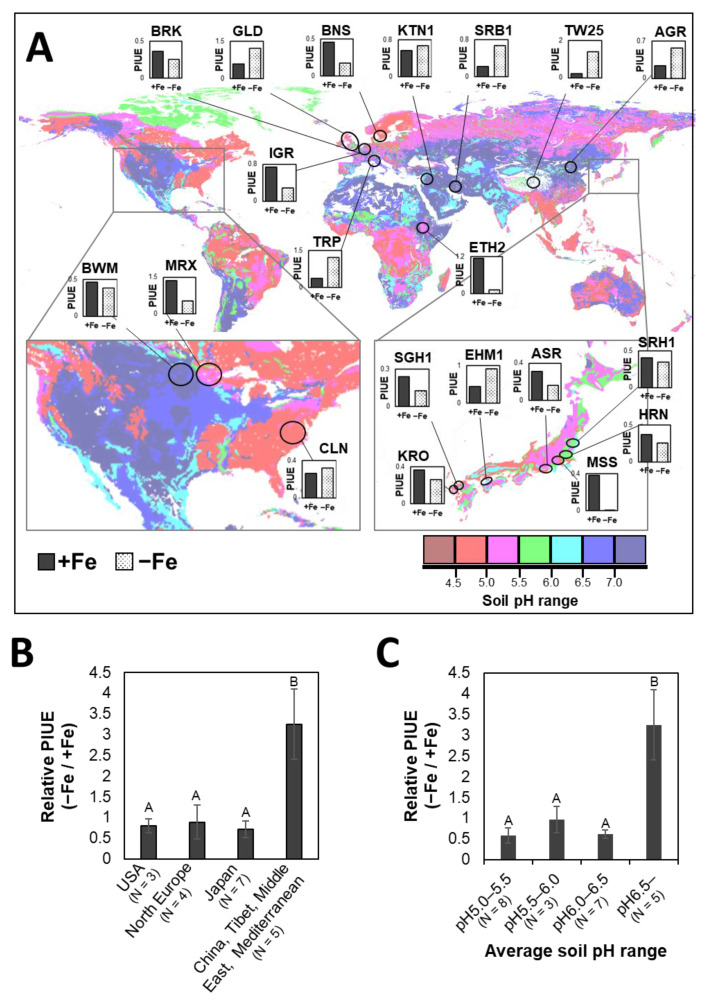
Relationship between soil pH and global distribution of barley varieties with a different calculated photosynthetic Fe-use efficiency (PIUE). The distribution of 20 barley varieties on the world soil pH map (**A**). The inserted small bar graphs in (**A**) represent the absolute PIUE (Figure 4A) of each variety in Fe-sufficient (+Fe) and Fe-deficient (−Fe) conditions. As shown in the color charts below the map, areas filled with dark blue indicate alkaline and with dark red indicate acidic subsoil pH (30–100 cm depth). The collection site or production site of each barley variety was placed on the map at its original location of cultivation. Comparison of relative PIUE as the ratio of Fe-deficiency to Fe-sufficiency classified by the different geographical areas among 20 barley cultivars (**B**) and by the soil pH of 23 barley varieties including wild barley *H. vulgare* spontaneum and *H. murinum* (**C**). The significant differences among the data in (**B**) or (**C**) were tested using the Tukey’s multiple tests (*p* < 0.05, same capital letters indicate no significant difference).The global soil pH map was created by a software Harmonized World Soil Database (HWSD) viewer (FAO/IIASA/ISRIC/ISS-CAS/JRC, 2012. Harmonized World Soil Database (version 1.2). FAO, Rome, Italy and IIASA, Luxemburg, Austria) [29].

**Figure 6 plants-10-00234-f006:**
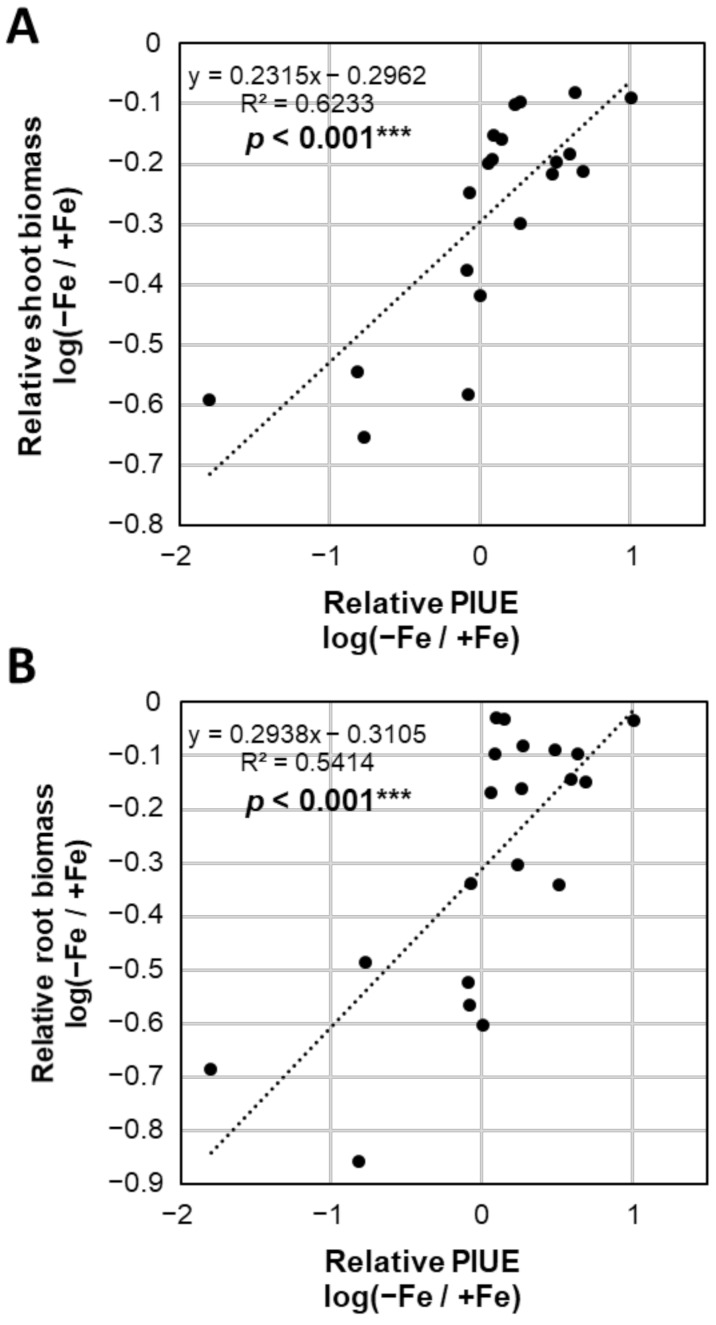
Correlation between biomass and photosynthetic Fe-use efficiency (PIUE) among barley varieties. Relative shoot biomass (**A**) and root biomass (**B**) as the ratio of Fe-deficiency to Fe-sufficiency. Plants in the vegetative growth phase were grown in hydroponic solution with or without Fe for 16 days. These variables (Appendix A) were converted to logarithms to fit the regression line. Significance of differences in Pearson’s correlation coefficient between relative PIUE and relative shoot biomass was determined using the Student’s *t* test (*** *p* < 0.001).

**Figure 7 plants-10-00234-f007:**
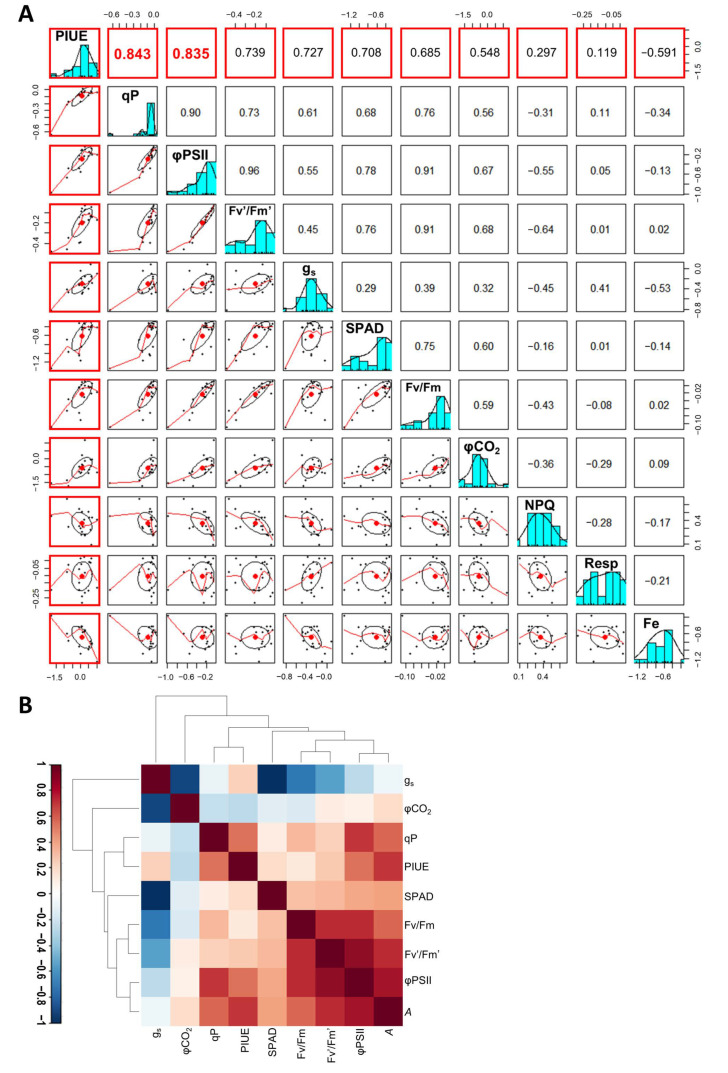
Correlation analysis of relative PIUE and relative variables among 18 barley varieties. (**A**) Scatter plot matrices among variables obtained in this study. These variables were the relative values as the ratio of Fe-deficiency to Fe-sufficiency and they were converted to logarithms. The values in the red square in the top row are Pearson correlation coefficients (r) as relative PIUE vs. relative qP, relative φPSII, relative Fv’/Fm’, relative stomatal conductance (g_s_), relative SPAD value, relative Fv/Fm, relative φCO_2_, relative respiration rate (Resp), relative NPQ, and relative leaf Fe concentration (Fe). The original data used for the analysis are presented in Appendix A. (**B**) Heatmap colored correlation matrix for 10 variables in the dataset of 18 barley varieties. Abbreviations: stomatal conductance (g_s_), quantum yield of carbon assimilation (φCO_2_), photochemical quenching coefficient (qP), photosynthetic Fe-use efficiency (PIUE), chlorophyll index (SPAD), maximal quantum yield of PSII photochemistry (Fv/Fm), quantum yield of PSII photochemistry (Fv’/Fm’), effective quantum yield of electron transport in the light-acclimated state (φPSII), net CO_2_ assimilation rate (*A*).

**Figure 8 plants-10-00234-f008:**
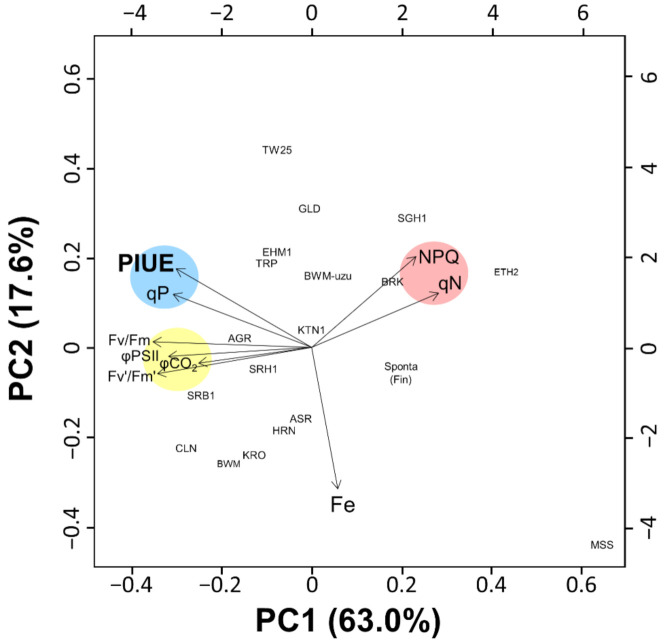
Principle component analysis (PCA) for nine variables in the dataset of 18 barley varieties. Blue, yellow, and red indicate clusters of photosynthetic Fe-use efficiency (PIUE)-related indices (PIUE and qP), electron transport-related indices (Fv/Fm, Fv’/Fm’, φPSII, and φCO_2_), and nonphotochemical quenching (qN and NPQ). Proportion of variances for PC1 and PC2 are shown in parentheses and in Appendix A. Loading of each variables is shown in Appendix A. Dataset of the logarithmic relative value [log(−Fe/+Fe)] of variables related to photosynthesis are listed in Appendix A. Abbreviations of plant variety names are summarized below: ‘Ehime Hadaka 1’ (EHM1), ‘Shiro Hadaka 1’ (SRH1), ‘Kairyo Ogara’ (KRO), ‘Haruna Nijo’ (HRN), ‘Akashinriki’ (ASR), ‘Saga Hadaka 1’ (SGH1), ‘Musashinomugi’ (MSS), ‘Colonial’ (CLN), ‘Bowman’ (BWM), ‘Bowman near-isogenic line *uzu1.a*’ (BWM-uzu), ‘Golden Promise’ (GLD), ‘Tripoli’ (TRP), ‘Sarab 1’ (SRB1), ‘Katana 1’ (KTN1), ‘Tibet White 25’ (TW25), ‘Ethiopia 2’ (ETH2), ‘Agriochriton’ (AGR), ‘Spontaneum’ originated from Finland (Sponta-Fin).

**Figure 9 plants-10-00234-f009:**
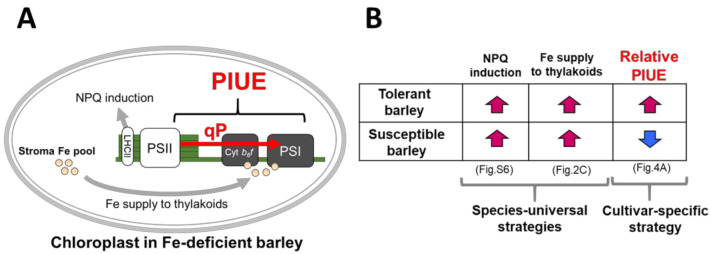
Model of adaptive mechanisms in chloroplasts of Fe-deficient barley. (**A**) Chloroplasts in Fe-deficient barley have at least three different mechanisms to adapt/acclimate to Fe-deficiency. High NPQ induction to protect from photoinhibition [13] and preferential Fe supply to the thylakoid membranes [17] are commonly observed in barley varieties grown under Fe-deficiency as fundamental acclimation mechanisms. Induction of high PIUE (the current study) was observed only in the barley varieties with Fe-deficiency-tolerance, which generally originated from regions covered with alkaline soils. (**B**) Relationship between Fe-deficiency-tolerance and the three adaptive mechanisms in chloroplasts. Red up-arrows and blue down-arrows indicate an increase and decrease (or non-induction) of each mechanism under Fe-deficiency. The universality of each mechanism within *H. vulgare* species is shown below the table. The data to reach this conclusion are shown in parentheses.

## Data Availability

Data is contained within the article or Appendix A.

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
