# Peer review of "Enhancement of Photosynthetic Iron-Use Efficiency Is an Important Trait of Hordeum vulgare for Adaptation of Photosystems to Iron Deficiency"

_plants, 2021, doi:10.3390/plants10020234_

Round 1

Reviewer 1 Report

The paper by Saito et al. describes a novel quantitative trait, the Photosynthetic Iron Use Efficiency index to characterise the response of barley varieties to iron deficiency using a multivariate approach. The authors have applied all necessary techniques to test their hypothesis and as a result the manuscript is one of the best scientific studies and at the same time one of the most exciting scientific readings I have read recently. I could not find a single spelling error or anything inappropriate in this work.

Congratulations!

Reviewer 2 Report

The authors claim that the tolerance to Fe deficiency in barley ecotypes is based likely in the optimization of electron flow from PSII to PSI.

Major comments:

1) The pictures in the figure 1C do not match with the SPAD values and fresh weight in the figures A and 1B. The authors must include pictures with the same frame, and similar brightness and contrast levels.

2) The results showed in the figure 2 are confused and extremely difficult to get clear conclusions. If the authors have problems with different SPAD values, the simple and best solution is decrease the lenght of Fe deficiency. But add different Fe levels and also modify Mn, Zn and Cu amount is not a correct experimental design.  

3) This manuscript is based only in Fe content but, under Fe deficency is very likely to observe a change in the amount of other macro and micro nutrients.

The authors must include the ICP values of all metals in the experiment.

4) Mn is the most important metal in the PSII activity and its deficiency cause alteration in many chlorophyll fluorescence parameters. qP value is related likely to PSI activity but the results showed in the manuscript are very indirect. Therefore, PSI activity must be measure under Fe deficiency.

5) In this manuscript, many correlations are associated to PIUE but, correlation is not causation

PSI and PIUE are correlated (based in qP only) but there is not any evidency how CO assimilation and Fe content in leaves is related to Fe-S cluster in PSI.

Correlation between PIUE and alkaline soils is flawed. The map is not good indicator of the soil pH in specific regions. Soil analysis in the area of barley ecotype origin must be a better indicator. Moreover, pH 6-7 is not consider too much alkaline to induce Fe defiency, as well as Mn and Zn.

6) Line 364-366. NPQ is a specific parameter of PSII but, the conclusions are based in a putative defect in the PSI activity under Fe deficincy. Therefore, the authors must explain how NPQ is related with Fe deficiency at molecular level.

7) Following the data of Table S2, the plant with lower PIUR values are also the plants with very low SPAD index, low stomatal conductance and biomass values, except BRK. Is possible that these plants are very affected by Fe deficiency and PIUE index is only and indicator of senescence leaves?

Long Fe deficiency cause chlorosis and oxidative stress in the plant. Why the authors decided a long Fe deficiency treatment 16-20 days?

Round 2

Reviewer 2 Report

 ok